# Does Visual Pretraining Help End-to-End Reasoning?

**Chen Sun**
Brown University, Google

**Calvin Luo**
Brown University

**Xingyi Zhou**
Google Research

**Anurag Arnab**
Google Research

**Cordelia Schmid**
Google Research

## Abstract

We aim to investigate whether end-to-end learning of visual reasoning can be achieved with general-purpose neural networks, with the help of visual pretraining. A positive result would refute the common belief that explicit visual abstraction (e.g. object detection) is essential for compositional generalization on visual reasoning, and confirm the feasibility of a neural network "generalist" to solve visual recognition and reasoning tasks. We propose a simple and general self-supervised framework which "compresses" each video frame into a small set of tokens with a transformer network, and reconstructs the remaining frames based on the compressed temporal context. To minimize the reconstruction loss, the network must learn a compact representation for each image, as well as capture temporal dynamics and object permanence from temporal context. We perform evaluation on two visual reasoning benchmarks, CATER and ACRE. We observe that pretraining is essential to achieve compositional generalization for end-to-end visual reasoning. Our proposed framework outperforms traditional supervised pretraining, including image classification and explicit object detection, by large margins.

## 1 Introduction

This paper investigates if an end-to-end trained neural network is able to solve challenging visual reasoning tasks [23, 65, 66] that involve inferring causal relationships, discovering object relations, and capturing temporal dynamics. A prominent approach [49] for visual reasoning is to construct a structured, interpretable representation from the visual inputs, and then apply symbolic programs [42] or neural networks [17] to solve the reasoning task. Despite their appealing properties, such as being interpretable and enabling the injection of expert knowledge, it is practically challenging to determine what kinds of symbolic representations to use and how to detect them reliably from visual data. In fact, the most suitable symbolic representation may differ significantly across different tasks: the representation for modeling a single human's kinematics (e.g. with body parts and joints) is unlikely to be the same as that for modeling group social behaviors (e.g. each pedestrian can be viewed as its own complete, independent entity). With the success of unified neural frameworks for multi-task learning [7], it is desirable to utilize a unified input interface (e.g. raw pixels) and to have the neural network learn to dynamically extract suitable representations for different tasks. However, how to learn a distributed representation with a deep neural network that *generalizes* similarly to learning methods based on symbolic representation [66] for visual reasoning remains an open problem.

The key hypothesis we make in this paper is that a general-purpose neural network, such as a Transformer [55], can be turned into an *implicit* visual concept learner with *self-supervised pretraining*. An implicit visual concept refers to a vector-based representation in an end-to-end neural network, which can be "finetuned" directly on the downstream tasks. Some of the learned implicit representations may be discretized into human-interpretable symbols for the purposes of human understanding of and feedback to the model. Others may correspond to part of, or a combination of

37th Conference on Neural Information Processing Systems (NeurIPS 2023).

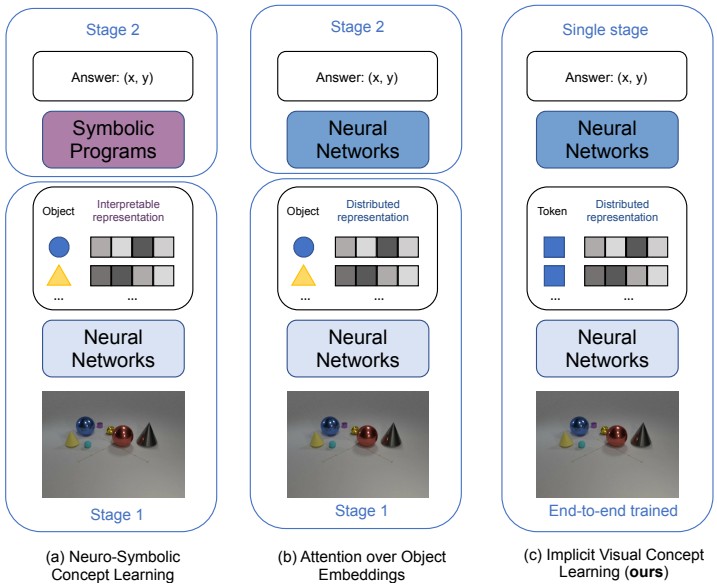

Figure 1: **Comparison between a neuro-symbolic approach, a hybrid approach with learned object embeddings [17], and our proposed approach for visual reasoning.** The illustration of each model family flows upwards, where visual inputs are encoded by neural networks (stage 1), and then processed by symbolic programs or another neural network to generate reasoning predictions (stage 2). Compared to (a) and (b), our approach does not require a separate "preprocessing" stage to extract the symbolic representation from visual inputs, and the self-supervised pretrained neural network can be end-to-end "finetuned" to the downstream visual reasoning tasks.

human-interpretable symbols. As opposed to explicit symbolic representation (e.g. object detection), implicit visual concepts do not require pre-defining a concept vocabulary or constructing concept classifiers, and also do not suffer from the early commitment or loss of information issues which may happen when visual inputs are converted into explicit symbols or frozen descriptors (e.g. via object detection and classification). A comparison between our approach and those that utilize explicit symbols under a pipeline-styled framework is visualized in Figure 1.

Our proposed representation learning framework, *implicit visual concept learner* (IV-CL) consists of two main components: first, a single image is *compressed* into a small set of tokens with a neural network. This is achieved by a vision transformer (ViT) network [19] with multiple "slot" tokens (e.g. the [CLS] token in ViT) that attend to the image inputs. Second, the slot tokens are provided as context information via a temporal transformer network for other images in the same video, where the goal is to perform video reconstruction via the *masked autoencoding* [27] objective with the temporal context. Despite its simplicity, the reconstruction objective motivates the emergence of two desired properties in the pretrained network: first, to provide context useful for video reconstruction, the image encoder must learn a compact representation of the scene with its slot tokens. Second, to utilize the context cues, the temporal transformer must learn to associate objects and their implicit representation across time, and also capture the notion of object permanence – the existence of an object even when it is occluded from the visual observations.

We conduct extensive ablation experiments on the Compositional Actions and TEmporal Reasoning (CATER) [23] benchmark and the Abstract Causal REasoning (ACRE) [66] benchmark. To better understand if and how end-to-end pretraining helps visual reasoning, we also consider the supervised pretraining paradigm, where the slot tokens in the Transformer network are pretrained to "decode" image-level labels or object locations and categories. Specifically, we adopt the Pix2Seq objective [13], which formulates object detection as an autoregressive "language" modeling task.

Our experimental results reveal the following observations: first, IV-CL learns powerful implicit representations that achieve competitive performance on CATER and ACRE, confirming that visual pretraining does help end-to-end reasoning. Second, the pretraining objective matters: networks pretrained on large-scale image classification benchmarks [15, 52] transfer poorly to the visual reasoning benchmarks, while object detection learns better representation for reasoning. However,

both are outperformed by IV-CL by large margins. Finally, we observe that the network inductive biases, such as the number of slot tokens per image, play an important role: on both datasets, we observe that learning a small number of slot tokens per image (1 for CATER and 4 for ACRE) lead to the best visual reasoning performance. To the best of our knowledge, our proposed framework is the first to achieve competitive performance on CATER and ACRE without the need to construct explicit symbolic representation from visual inputs.

In summary, our paper makes the following two main contributions: First, unlike common assumptions made by neuro-symbolic approaches, we demonstrate that compositional generalization for visual reasoning can be achieved with end-to-end neural networks and self-supervised visual pretraining. Second, we propose IV-CL, a self-supervised representation learning framework, and validate its effectiveness on the challenging CATER and ACRE visual reasoning benchmarks against supervised visual pretraining counterparts. Code and checkpoints will be released.

## 2 Related Work

**Neural Network Pretraining.** Huge progress has been made towards building unified learning frameworks for a wide range of tasks, including natural language understanding [16, 48, 8, 40], visual recognition [36, 35, 63, 22], and multimodal perception [33, 50, 38, 24, 3]. Unfortunately, most of the "foundation models" [7] for visual data focus on perception tasks, such as object classification, detection, or image captioning. Despite improved empirical performance on the visual question answering task [32, 64], visual reasoning remains challenging when measured on more controlled benchmarks that require compositional generalization and causal learning [66, 23, 14]. It is commonly believed that symbolic or neurosymbolic methods [42, 62, 37, 4], as opposed to general-purpose neural networks, are required to achieve generalizable visual reasoning [61, 66, 65]. To our knowledge, our proposed framework is the first to demonstrate the effectiveness of a general-purpose end-to-end neural network on these visual reasoning benchmarks.

**Self-supervised Learning from Images and Videos.** Self-supervised learning methods aim to learn strong visual representations from unlabelled datasets using pre-text tasks. Pre-text tasks were initially hand-designed to incorporate visual priors [18, 69, 10]. Subsequent works used contrastive formulations which encourage different augmented views of the same input to map to the same feature representation, whilst preventing the model from collapsing to trivial solutions [45, 12, 28, 26, 2]. One challenge of the contrastive formulation is the construction of positive and negative views, which has been shown to critically impact the learned representation [12, 59, 51]. Whereas contrastively learned representations may not easily transfer across domains [46], our pretraining successfully generalizes to visually different datasets, such as from ACRE to RAVEN.

Our work is most related to masked self-supervised approaches. Early works in this area used stacked autoencoders [56] or inpainting tasks [47] with convolutional networks. These approaches have seen a resurgence recently, inspired by BERT [16] and vision transformers [19]. BEiT [6] encodes masked patches with discrete variational autoencoders and predicts these tokens. Masked Autoencoders (MAE) [27], on the other hand, simply regress to the pixel values of these tokens. MAE has been extended to regress features [57] and to learn video representations [53, 20]. Our training objective is different, as it is predictive coding based on compressed video observations. We confirm empirically that the proposed method outperforms MAE and its video extension by large margins.

**Object-centric Representation for Reasoning.** Most of the existing neuro-symbolic [42, 62] and neural network [17] based visual reasoning frameworks require a "preprocessing" stage of symbolic representation construction, which often involves detecting and classifying objects and their attributes from image or video inputs. Our proposed framework aims to investigate the effectiveness of single-stage, end-to-end neural networks for visual reasoning, which is often more desirable than the two-stage frameworks for scenarios that require transfer learning or multi-task learning. In order to obtain the object-centric, or symbolic representation in the preprocessing stage, one can rely on a supervised object detector [42, 54], such as Mask R-CNN [29]. An alternative approach is to employ self-supervised objectives and learn low-level features that are correlated with objects, such as textures [21, 30, 44], or objects themselves [9, 41, 11]. In practice, supervised or self-supervised approaches for object detection and object-centric representation learning may suffer from the lack of supervised annotations, or from noisy object detection results. For example, it was previously observed that object-centric representations are beneficial for transfer learning to temporal event classification only when ground truth object detections are used for training and evaluation [68].

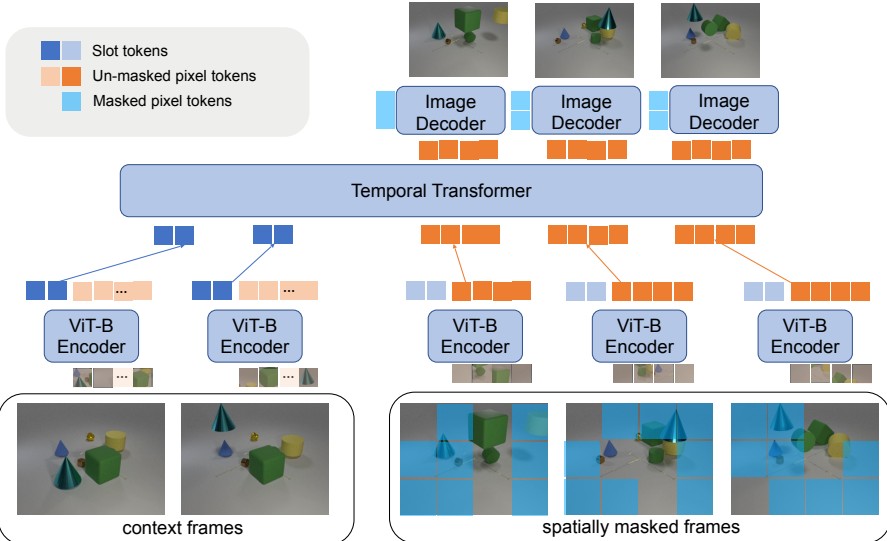

Figure 2: **IV-CL self-supervised pretraining.** We consider the video reconstruction objective via masked autoencoding: A ViT-B image encoder is tasked to (1) extract visual representations (orange) for the unmasked patches per image and (2) compress an image into a small set of slot tokens (blue). A temporal transformer then propagates the information from the slot representations and patch-level representations from neighboring frames, which are essential for successful reconstruction.

## 3 Method

We now introduce the proposed implicit visual concept learning (IV-CL) framework. We follow the pretraining and transfer learning paradigm: during pretraining (Figure 2), we task a shared image encoder to output patch-level visual embeddings along with a small set of slot tokens that compress the image's information. The pretraining objective is inspired by masked autoencoding (MAE) for unlabeled video frames, where the aim is to reconstruct a subset of "masked" image patches given the "unmasked" image patches as context. Compared to the standard MAE for images [27], the image decoder has access to two additional types of context information: (1) The encoded patch embedding from the unmasked image patches of the neighboring frames; (2) The encoded slot tokens from a subset of context frames. The context information is encoded and propagated by a temporal transformer network. To successfully reconstruct a masked frame, the image encoder must learn a compact representation of the full image via the slot tokens, and the temporal transformer has to learn to capture object permanence and temporal dynamics.

After pretraining, the image decoder is discarded, and only the image encoder and temporal transformer are kept for downstream visual reasoning tasks. The inputs to the temporal transformer are the slot tokens encoded from individual, unmasked video frames. We use the full finetuning strategy where the weights of both the newly added task decoder (e.g. a linear classifier), and the pretrained image and temporal transformers are updated during transfer learning.

**Image Encoder:** We adopt the Vision Transformer (ViT) backbone to encode each image independently: an input image is broken into non-overlapping patches of $16\times16$ pixels, which are then linearly projected into patch embeddings as inputs to the transformer encoder. Spatial information is preserved by sinusoidal positional encodings. We use the standard ViT-Base configuration which has 12 Transformer encoder layers. Each layer has hidden size of 768, MLP projection size of 3072, and 12 attention heads. During pretraining, a subset of video frames are spatially masked randomly given a masking ratio, only the unmasked image patches are fed into the ViT-B encoder. For context frames and during transfer learning, all image patches are provided as inputs to the image encoder.

**Slot Tokens:** In the seminal work by Locatello et al. [41], slot tokens are defined as soft cluster centroids that group image pixels, where the goal is unsupervised object detection. Each slot token repeatedly attends to the raw image inputs and is iteratively refined with a GRU network. We borrow their terminology, and use slots to denote the representational bottleneck in which we hope to encode implicit visual concepts, such as object-centric information. We generalize their slot update rules

by: (1) iteratively updating the visual representation with layers of the Transformer encoder (ViT); (2) replacing cross-attention with multi-headed self-attention; (3) using MLP layers with untied weights to update the intermediate slot representation as opposed to a shared GRU network. These two modifications allow us to implement "slot attention" directly with a Transformer encoder, simply by prepending slot tokens as additional inputs to the encoder (similar to `[CLS]` tokens). The initial slot embeddings at the input of the visual encoder are implemented as a learnable embedding lookup table. To compare the effectiveness of different methods to aggregate "slot" information, we also explore single-headed soft attention and Gumbel-max attention as used by [60].

**Temporal Transformer:** To propagate temporal information across frames, we use another transformer encoder (with fewer layers than the ViT-B image encoder) which takes the tokens encoded by the image encoder as its inputs. During pretraining, the slot tokens from context frames, along with the unmasked patch tokens from the query frames are concatenated together and fed into the temporal transformer. For each query image, the temporal transformer outputs its corresponding unmasked patch tokens *contextualized* from both the unmasked patches from neighboring query frames and the slot tokens from context frames. The contextualized patches are then fed into the image decoder to compute the reconstruction loss. To preserve temporal position information, we use learned positional embeddings (implemented with an embedding lookup table). When finetuned on a reasoning task, the temporal transformer takes the slot tokens encoded by the image encoder as its inputs. Putting the image encoder and the temporal transformer together, the overall video encoder used for finetuning can be viewed as a factorized space-time encoder proposed by [5]. It is more parameter-efficient than the vanilla video transformer used by [53].

**Image Decoder for Pre-training:** We use the same image decoder as in [27]. The query images are decoded independently given the contextualized unmasked patch tokens. The image decoder is implemented with another transformer, where masked patch tokens are appended to the contextualized unmasked patch tokens as inputs to the image decoder. Sinusoidal positional encodings are used to indicate the spatial locations of individual patch tokens. We use the same number of layers, hidden size, and other hyperparameters as recommended by [27]. During pre-training, we use mean squared error to measure the distance between the original query image patches and the reconstructed patches.

**Transfer Learning:** As the goal of pre-training is to learn the slot tokens which we hope to compress an input image into a compact set of implicit visual concept tokens, we only ask the image encoder to generate the slot tokens during finetuning, which are fed to the temporal transformer as its inputs. We then average pool the output tokens of the temporal transformer and add a task-specific decoder to make predictions. Both benchmarks used in our experiments can be formulated as multi-class classification: for CATER, the goal is to predict the final location of the golden snitch, where the location is quantized into one of the $6 \times 6$ positions; in ACRE, the goal is to predict whether the platform is activated, unactivated, or undetermined given a query scenario. We use linear classifiers as the task-specific decoders with standard softmax cross-entropy for transfer learning.

**Supervised Pretraining Baselines:** To better understand if visual pretraining helps end-to-end reasoning, we consider two types of supervised pretraining baselines. The first is the "classical" image classification pretraining which often exhibits scaling laws [52] when transferred to other visual recognition benchmarks. The second is the object detection task, which intuitively may also encourage the emergence of object-centric representations (per task requirement) inside the neural network. Both pretraining objectives can be directly applied on the same Transformer architecture as utilized in IV-CL, with different designs on the task specific decoders (which are discarded for visual reasoning finetuning). For image classification, we directly treat the slot token as a `[CLS]` token and add a linear classifier on top of it. For object detection, to make minimal modification to our framework, we follow the design proposed by Pix2Seq [13], which parameterizes the bounding box annotations as discrete tokens, and formulates the training objective as an autoregressive sequence completion task. The inputs to the autoregressive decoder are the encoded slot tokens. We adopt the same sequence construction and augmentation strategies as in Pix2Seq.

## 4 Experiments

### 4.1 Experimental Setup

**Benchmarks:** In the classic "shell game", a ball is placed under a cup and shuffled with other empty cups on a flat surface; then, the objective is to determine which cup contains the ball. Inspired by

this, CATER is a dataset composed of videos of moving and interacting CLEVR [34] objects. A special golden ball, called the "snitch", is present in each video, and the associated reasoning task is to determine the snitch's position at the final frame. Solving this task is complicated by the fact that larger objects can visually occlude smaller ones, and certain objects can be picked up and placed down to explicitly cover other objects; when an object is covered, it changes position in consistence with the larger object that covers it. In order to solve the task, a model must learn to reason not only about objects and movement, but also about object permanence, long-term occlusions, and recursive covering relationships. Each video has 300 frames, and we use the static camera split for evaluation.

The ACRE dataset tests a model's ability to understand and discover causal relationships. The construction of the dataset is motivated by the Blicket experiment in developmental psychology [25], where there is a platform as well as many distinct objects, some of which contain the "Blicketness" property. When at least one object with the "Blicketness" property is placed on the platform, music will be played; otherwise, the platform will remain silent. In ACRE, the platform is represented by a large pink block that either glows or remains dim depending on the combination of CLEVR objects placed on it. Given six evidence frames of objects placed on the platform, the objective of the reasoning task is to determine the effect a query frame, containing a potentially novel object combination, would have on the platform. Possible answers including activating it, keeping in inactive, or indeterminable.

**Pretraining data:** We use the unlabeled videos from the training and validation splits of the CATER dataset for pretraining. Both the static and moving camera splits are used, which contains 9,304 videos in total. In our experiments, we observe that ACRE requires higher resolution inputs during pretraining and finetuning. Our default preprocessing setup is to randomly sample 32 frames of size $64 \times 64$ for pretraining the checkpoints that are transferred to CATER, and 16 frames of size $224 \times 224$ for pretraining the checkpoints that are transferred to ACRE. The randomly sampled frames are sorted to preserve the arrow of time information. No additional data augmentations are performed.

**Transfer learning:** For CATER, we evaluate on the static split which has 3,065 training, 768 validation, and 1645 test examples. We select the hyperparameters based on the validation performance, then use both training and validation data to train the model to be evaluated on the test split. By default, we use 100 randomly sampled frames of size $64 \times 64$ during training, and 100 uniformly sampled frames of stride 3 during evaluation. For ACRE, we explore all three splits, all of which contain 24,000 training, 8,000 validation, and 8,000 test examples. We use the validation set to select hyperparameters and use both training and validation to obtain the models evaluated on the test split.

**Default hyperparameters:** We use the Adam optimizer for pretraining with a learning rate of $10^{-3}$, and the AdamW optimizer for transfer learning with a learning rate of $5 \times 10^{-5}$. The pretraining checkpoints are trained from scratch for 1,000 epochs using a batch size of 256. For transfer learning, we finetune the pretrained checkpoints for 500 epochs using a batch size of 512. All experiments are performed on TPU with 32 cores. Below we study the impact of several key model hyperparameters.

## 4.2 IV-CL vs. Supervised Pretraining

We first compare our proposed IV-CL to traditional supervised pretraining on both detection and classification tasks. For classification, we consider the same ViT-B visual encoder trained on ImageNet-21K [15] and JFT [52]. For object detection, we consider an in-domain object detection benchmark dataset called LA-CATER [49]. LA-CATER matches the visual appearance of CATER; it was created to study the benefit of modeling object permanence and provides frame-level bounding box annotations for all visible and *occluded* objects. We validated the correctness of our object detector on the COCO benchmark, which achieves comparable performance to the original Pix2Seq implementation. On the LA-CATER validation set, we observe 82.4% average precision (AP) at an IOU threshold of 50%. Whereas one might expect almost perfect performance on such a synthetic environment, this can be explained by the inherent properties of the dataset; frame-level object detection on LA-CATER also evaluates the detection of occluded and invisible objects, which is indeterminable when given only single, static images as inputs. We also consider a classification pretraining baseline to count the number of unique objects in LA-CATER frames.

We note three remarkable trends when inspecting the results in Table 1. First, we observe that none of the models pretrained with supervision outperforms their self-supervised counterpart. Instead, their performance on both CATER and ACRE fall behind IV-CL by large margins. Second, when

Table 1: **Self-supervised visual pretraining vs. supervised pretraining.** We compare our proposed pretraining with traditional supervised classification or detection pretraining.

| Objective | Pretrain data | CATER | ACRE (comp) |
|---|---|---|---|
| Random Init. | - | 3.34% | 38.78% |
| Detection | LA-CATER | 56.64% | 67.27% |
| Classification | LA-CATER | 41.48% | 64.78% |
| Classification | ImageNet-21k | 55.58% | 60.73% |
| Classification | JFT | 54.07% | 48.32% |
| IV-CL | CATER | **70.14** (±0.59)% | **93.27** (±0.22)% |

comparing the detection and classification objectives, we observe that the detection pretraining outperforms classification pretraining significantly. This can be potentially explained by the domain gap between natural image datasets and CLEVR-style datasets, or that object detection objective encourages the learning of object-centric representations in the slot tokens. To better understand this, we perform addition ablations by replacing the object detection dataset with COCO [39], which is a natural image dataset. We observe similar transfer learning performance as LA-CATER pretraining. Additionally, we perform a probing experiment where we ask the object detection decoder to make predictions with a single randomly sampled slot token. We empirically observe that each token appears to focus on one or a small subsets of the objects in the scene, and different tokens are complementary to each other. Both observations indicate that the stronger performance of object detection pretraining is likely due to the "object-centric" objective itself. Finally, we observe a counterexample of the "scaling law": larger scale classification pretraining (JFT) leads to significantly worse performance than smaller scale pretraining (ImageNet-21k).

### 4.3 Visualizations of the Learned Slots

To help understand what visual concepts are implicitly captured by IV-CL, we visualize the attention heatmaps from each learned slot token back to the image pixels. This is implemented with the attention rollout technique [1]. Figure 3 shows examples of the attention heatmaps after (a) self-supervised pretraining on CATER, and after (b) finetuning for visual reasoning on ACRE.

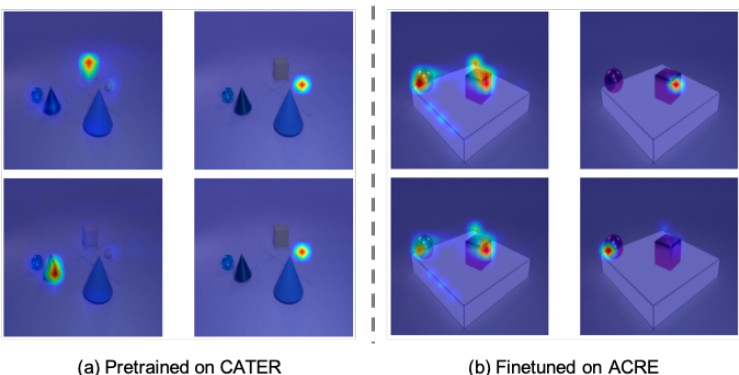

(a) Pretrained on CATER      (b) Finetuned on ACRE

Figure 3: **Visualization of 4 slots of an IV-CL model after pretraining on CATER (left) and finetuning on ACRE (right)**. Each heatmap is generated by attention rollout [1] to the input pixels. A brighter color indicates higher attention weight.

We observe two general patterns by inspecting the pretrained slot heatmaps: first, a subset of the heatmaps exhibit object-centric behavior, as in Figure 3(a). Each slot tends to focus on an individual object, and most of the objects in an image are covered by combining the heatmaps from all four slots. However, we also observe that sometimes the slots are not completely disentangled with respect to individual objects, which indicates that the implicit representations obtained after IV-CL pretraining do not learn perfectly disentangled visual concepts, and further finetuning is necessary to achieve compositional generalization on visual reasoning. We then inspect the heatmaps after finetuning for visual reasoning on ACRE in Figure 3(b). We observe some slots model relationships among objects

Table 2: **CATER Pretraining** with different mask ratios, context sizes, and frame lengths.

| (a) Mask ratio | | (b) Context size | | (c) Frame length | |
|---|---|---|---|---|---|
| Ratio | Acc. | Size | Acc. | Length | Acc. |
| 37.5% | **70.14%** | 8 | **70.14%** | 32 | **70.14%** |
| 12.5% | 66.35% | 0 | 65.35% | 8 | 62.28% |
| 50% | 66.57% | 4 | 67.47% | 16 | 66.63% |
| 87.5% | 61.94% | 16 | 64.34% | 64 | 68.25% |

Table 3: **CATER Pretraining** with different number of slots, and pooling strategies

| (a) Number of slots | | (b) Where to pool | | (c) How to pool | |
|---|---|---|---|---|---|
| # slots | Acc. | Layer | Acc. | Method | Acc. |
| 1 | **70.14%** | 11 | **70.14%** | Slice | **70.14%** |
| 2 | 66.52% | 5 | 55.80% | Soft | 64.23% |
| 8 | 64.45% | 9 | 68.86% | Hard | 65.90% |

and the platform, and some focus on individual objects. Intuitively, both types of information are needed to solve the ACRE benchmark. Finally, we visualized the attention heatmap of a ImageNet-21k pretrained model after finetuning on ACRE. We observe that the heatmaps often "collapse" on a small subset of the same objects, which coincides with its lower reasoning performance.

## 4.4 Ablation Study

Next, we ablate our key design choices. We present our ablation study on CATER in Table 2.

**Masking ratio:** Contrary to the large masking ratio (75%) employed in vanilla MAE [27], we found that the optimal masking ratio was 37.5% on downstream CATER accuracy. This is perhaps due to the fact that CATER is designed to test "compositional generalization", and so the spatial context provides less information than in natural images and video.

**Number of Total Frames and Context Frames:** We also study the impact of the number of frames IV-CL is pretrained on, and find the best performance on 32 frames. Fixing the total number of pretraining frames, we then ablate over the number of context frames, which are the frames from which slot representations are generated. When no context frame is used, we essentially utilize only patch-level representations to perform reconstruction with the temporal transformer (simulating a per-frame MAE followed by a temporal transformer). We find that the best performance is achieved with 8 context frames, which balances the number of slot representations with patch-level representations.

**Number of Slot Tokens:** Another useful ablation is on the impact of the number of slots used for CATER and ACRE. In CATER, we find that only 1 slot token per frame is enough to solve the reasoning task. We believe that this may be due to how the reasoning objective of CATER is designed: to successfully perform snitch localization, the model need only maintain an accurate prediction of where the snitch actually or potentially is, and can ignore more detailed representation of other objects in the scene. Under the hypothesis that the slot tokens represent symbols, perhaps the singular slot token is enough to contain the snitch location. On the other hand, when ablating over the number of tokens for the ACRE task (see Appendix), we find that a higher number of tokens is beneficial for reasoning performance. This can potentially be explained by the need to model multiple objects across evidence frames in order to solve the final query; under our belief that slot tokens are encoding symbols, multiple may be needed in order to achieve the best final performance.

**Slot Pooling Layer and Method:** We ablate over which layer to pool over to generate the slot tokens. The patch tokens are discarded after the pooling layer, and only the slot tokens are further processed by the additional Transformer encoder layers. As expected, it is desirable to use all image encoder layers to process both slot and patch tokens. Additionally, we also study the impact of slot pooling method, and observe that adding additional single-headed soft attention and Gumbel-max attention are outperformed by simply using the slot tokens directly.

Table 4: **Results on CATER (static).** IV-CL performs the best among non-object-centric methods, and performs competitively with methods with object-supervision.

| Method | Object-centric | Object superv. | Top-1 Acc. (%) | Top-5 Acc. (%) |
|---|---|---|---|---|
| OPNet [49] | ✓ | ✓ | **74.8** | - |
| Hopper [70] | ✓ | ✓ | 73.2 | **93.8** |
| ALOE [17] | ✓ | ✗ | 70.6 | 93.0 |
| Random Init. | ✗ | ✗ | 3.3 | 18.0 |
| MAE (Image) [27] | ✗ | ✗ | 27.1 | 47.8 |
| MAE (Video) | ✗ | ✗ | 63.7 | 82.8 |
| IV-CL (ours) | ✗ | ✗ | **70.1** $\pm$ 0.6 | **88.3** $\pm$ 0.2 |

Table 5: **Results on ACRE compositionality, systematicity, and I.I.D. splits.** IV-CL performs the best among all methods on the compositionality split, and performs competitively on other splits.

| Method | Object-centric | Object superv. | comp (%) | sys (%) | iid (%) |
|---|---|---|---|---|---|
| NS-OPT [66] | ✓ | ✓ | 69.04 | 67.44 | 66.29 |
| ALOE [17] | ✓ | ✗ | **91.76** | **93.90** | - |
| Random Init. | ✗ | ✗ | 38.78 | 38.57 | 38.67 |
| MAE (Image) [27] | ✗ | ✗ | 80.27 | 76.32 | 80.81 |
| MAE (Video) | ✗ | ✗ | 78.85 | 71.69 | 77.14 |
| IV-CL (ours) | ✗ | ✗ | **93.27** $\pm$ 0.22 | **92.64** $\pm$ 0.30 | **92.98** $\pm$ 0.80 |

## 4.5 Comparison with State-of-the-Art

We compare our IV-CL framework with previously published results. As most of the prior work require explicit object detection and are not end-to-end trained, we reimplement an image-based MAE [27] and a video-based MAE [53] baseline and analyze the impact of inductive biases (using slot tokens or not) as well as pretraining objectives (predictive coding given compressed context, or autoencoding the original inputs) on the reasoning performance. Our reimplementation of image and video MAEs achieve very similar performances on their original benchmarks. However, for video-based MAE, we observe that the "un-factorized" backbone leads to training collapse on CATER. We hence adjust the backbone to be "factorized" as illustrated in Figure 2. We follow the same pretraining and hyperparameter selection procedures as for IV-CL.

Table 4 compares the result of IV-CL against other state-of-the-art models on CATER. We also compare IV-CL on ACRE against other existing models in Table 5. We cite the comparable results reported by the original authors when available. IV-CL achieves the best performance among the approaches that do not depend on explicit object-centric representation, and overall state-of-the-art performance on ACRE.

## 4.6 Transfer Learning to RAVEN

We explore generalization to a visually different reasoning benchmark, RAVEN [65]. Inspired by Raven's Progressive Matrices (RPM), its goal is to evaluate a machine learning model's structural, relational, and analogical reasoning capabilities. The reasoning task is to determine which of eight candidate geometrical figures naturally follow the patterned sequence of eight context figures. We explore all seven reasoning scenarios and perform finetuning on all training and validation examples (56,000 examples). The pretraining and finetuning hyperparameters exactly match those for ACRE, but the model now takes in 16 images as input (8 for context, 8 for answers). We report generalization performance on RAVEN in Table 6. We observe that the pretrained representation is generalizable, as IV-CL achieves competitive performance on RAVEN [65] with the same pretrained model and finetuning hyperparameters as ACRE, despite the different visual appearances across the datasets.

Table 6: **Transfer learning results on RAVEN.** We follow the same pretrained representation and finetuning hyperparameters as for ACRE.

| Method | Average | Center | 2×2 Grid | 3×3 Grid | L-R | U-D | O-IC | O-IG |
|---|---|---|---|---|---|---|---|---|
| LSTM | 13.1 | 13.2 | 14.1 | 13.7 | 12.8 | 12.4 | 12.2 | 13.0 |
| ResNet + DRT [65] | 59.6 | 58.1 | 46.5 | 50.4 | 65.8 | 67.1 | 69.1 | 60.1 |
| CoPINet [67] | 91.4 | 95.1 | 77.5 | 78.9 | **99.1** | **99.7** | **98.5** | 91.4 |
| SCL [58] | 91.6 | 98.1 | **91.0** | **82.5** | 96.8 | 96.5 | 96.0 | 80.1 |
| IV-CL (ours) | **92.5** | **98.4** | 82.6 | 78.4 | 96.6 | 97.2 | 99.0 | **95.4** |

## 4.7 Generalization to Real Videos

Finally, we attempt to answer the question: Would our proposed IV-CL framework work on real videos? We consider the Something-Else benchmark [43], which consists of short videos capturing the interactions between human hands and different objects. This benchmark focuses on relational reasoning, especially on compositional generalization across different object categories. We consider the base split and the "compositional" split. The base split contains 112,397 training videos and 12,467 validation videos, across 88 categories. The compositional split contains 54,919 training videos and 57,876 validation videos, across 174 categories. Each category corresponds to a fine-grained activity that requires spatiotemporal relation reasoning. The compositional split is designed to include disjoint object types for each category between the training set and the validation set.

Table 7: **Performance Evaluation on Something-Else.** We consider the base and compositional splits. *: Uses groundtruth box annotations during evaluation.

| Model | Split | Object Supervision | Top-1 Acc. (%) | Top-5 Acc. (%) |
|---|---|---|---|---|
| STRG, STIN+OIE+NL [43] | Base | ✓ | 78.1 | 94.5 |
| ORViT [31]* | Base | ✓ | 87.1 | 97.6 |
| IV-CL (Ours) | Base | ✗ | 79.1 | 95.7 |
| STRG, STIN+OIE+NL [43] | Comp | ✓ | 56.2 | 81.3 |
| ORViT [31]* | Comp | ✓ | 69.7 | 90.1 |
| IV-CL (Ours) | Comp | ✗ | 59.6 | 85.6 |

Due to the large domain gap between CATER and Something-Else videos, we choose to perform pretraining directly on the corresponding training splits of the Something-Else benchmark. We use the same pretraining and finetuning hyper parameters as in ACRE, except that we use 16 frames sampled at stride size of 2 during finetuning. During both pretraining and finetuning, we apply the standard video data augmentation techniques as used by prior work (e.g. [5]). In Table 7, we observe that our method generalizes well to real videos, and it achieves competitive performance compared to methods that use annotated boxes during training (STRG, STIN+OIE+NL) and evaluation (ORViT).

## 5 Conclusion and Future Work

In this work we demonstrate that competitive visual reasoning can be achieved in a general-purpose end-to-end neural network, with the help of self-supervised visual pretraining. Our proposed implicit visual concept learner (IV-CL) framework leverages a Transformer encoder to "compress" visual inputs into slot tokens, and is trained with a self-supervised video reconstruction objective. Quantitative and qualitative evaluations confirm the effectiveness of IV-CL on CATER and ACRE visual reasoning benchmarks, when compared to supervised visual pretraining and neuro-symbolic approaches. A limitation of our work is that evaluations are performed purely on synthetic reasoning tasks. We believe extending evaluation to large-scale natural video reasoning benchmarks, building a joint model for visual recognition and reasoning, and exploring how to incorporate explicit object-centric knowledge when such knowledge is available are interesting future directions to pursue.

**Acknowledgements:** Part of the research was conducted while C.L. worked as a student researcher at Google. C.S. and C.L. are in part supported by research grants from Honda Research Institute, Meta AI, Samsung Advanced Institute of Technology, and a Richard B. Salomon Faculty Research Award.

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

# 6   Additional Experimental Details

**Transfer Learning Framework.** In Figure 2, we visualized our proposed self-supervised pretraining framework. Once the representation network has been pretrained, we discard the image decoder and only use the ViT-B image encoder, along with the pretrained temporal transformer. An illustration of the transfer learning process is shown in Figure A1.

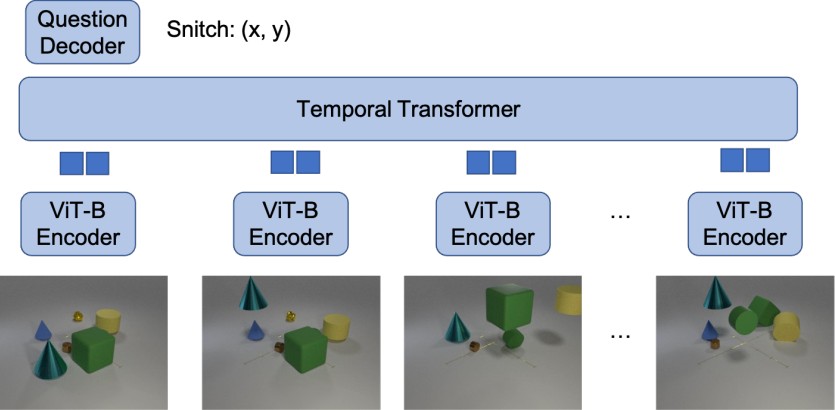

Figure A1: **An illustration of the transfer learning process.** Both the ViT-B image encoder and the temporal transformer are transferred to downstream visual reasoning tasks to encode video inputs. Unlike pretraining, only the slot tokens are provided as inputs to the temporal transformer.

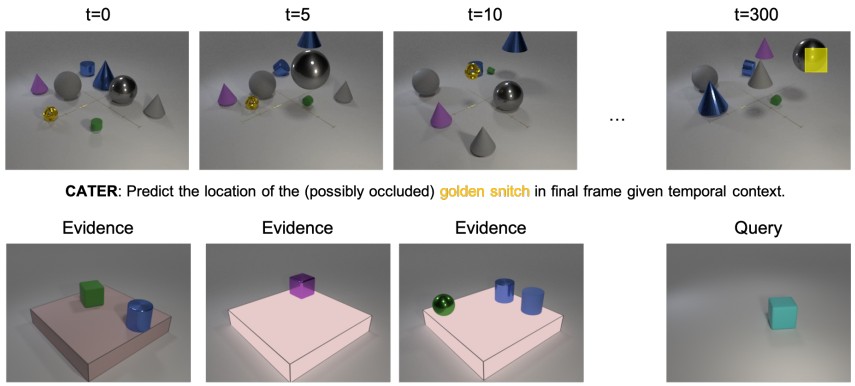

Figure A2: **Illustration of the CATER (top) and ACRE (bottom) benchmarks.**

**Illustration of the Benchmarks.** In Figure A2, we provide the illustrations of the CATER benchmark and the ACRE benchmark, respectively. The CATER benchmark features a special golden ball, called the "snitch", and the associated reasoning task is to determine the snitch's position at the final frame despite occlusions. Object locations in the CATER dataset are denoted by positions on an invisible 6-by-6 grid; therefore, in essence, the CATER task boils down to a 36-way classification problem. The CATER dataset features a split where the camera is statically fixed to a particular angle and position throughout the videos, as well as a moving camera split where the viewing angle is able to change over time. We use the static split for evaluation. Each video has 300 frames.

The ACRE benchmark is inspired by the research on developmental psychology: Given a few context demonstrations of different object combinations, as well as the resulting effect, young children have been shown to successfully infer which objects contain the "Blicketness" property, and which combinations would cause the platform to play music. ACRE explicitly evaluates four kinds of reasoning capabilities: direct, indirect, screened-off, and backward-blocking. Having the query frame be a combination that was explicitly provided in the context frames tests a model's direct reasoning ability. Indirect reasoning can be tested by a novel query combination, the effect of which

requires understanding multiple context frames to deduce. In screen-off questions, the model must understand that as long as a singular Blicket object is placed on the platform, the entire combination would cause it to light up. In backward-blocking questions, the model must recognize when the effect of a query combination cannot be determined from the provided context frames. Furthermore, ACRE features three different dataset splits to test model generalization: Independent and Identically Distributed (I.I.D.), compositionality (comp), systematicity (sys). In the compositionality split, the shape-material-color combinations of the CLEVR objects in the test set are not seen before in the train split; therefore, the model must learn to generalize across object attributes. In the systematicity split, the evidence frames of the train split contain three lit up examples, whereas the evidence frames of the test split contain four.

**Number of the Slot Tokens.** In Table A1, we provide ablation experiment on the impact of the number of slot tokens for the reasoning performance on all splits. Unlike CATER, whose goal is to infer the position of a single object, the "snitch", the ACRE benchmark requires reasoning over combinations of objects, and their relationship with the platform. As a result, we generally observe that more slot tokens are needed to achieve optimal performance. We observe that the performance starts to saturate given four or eight slots.

Table A1: **ACRE # tokens**. We show results on compositionality (comp), systematicity (sys), and I.I.D. (iid) splits.

| # slots | comp | sys | iid |
|---------|--------|--------|--------|
| 1 | 91.75% | 90.34% | 90.96% |
| 2 | 90.82% | 88.21% | 88.73% |
| 4 | 93.27% | **92.64%** | **92.98%** |
| 8 | **95.54%** | 86.18% | 88.97% |
| 64 | 90.45% | 80.07% | 90.82% |

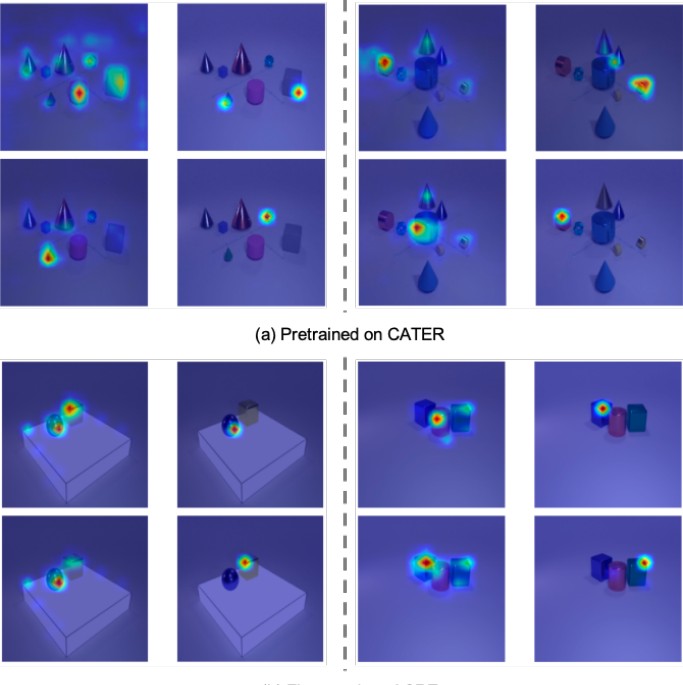

(a) Pretrained on CATER

(b) Finetuned on ACRE

Figure A3: **Visualizations of the Slot Tokens.** The top row corresponds to the attention heatmaps from the slot tokens after pretraining on CATER, and the bottom row corresponds to the heatmaps after finetuning on ACRE.

**Visualizations of Slot Tokens.** Figure A3 provides additional visualizations of the slot token attention heatmaps after pretraining on CATER, and finetuning on ACRE, respectively. We follow the same attention rollout technique as in Figure 3. For ACRE, we show the example when the platform is visible (context information) on bottom left, and when the platform is invisible (question) on the bottom right. We observe a consistent trend that a subset of the heatmaps exhibit object-centric behavior, especially before finetuning on ACRE. After finetuning, we observe that some slots remain focusing on individual objects, while the others attempt to model the relationships among different objects and the platform.

**Our MAE baselines** are pretrained with the same hyper parameters (e.g. optimization and mask ratio) as IV-CL, which we have observed to be optimal based on the validation set performance. The image encoders for all methods are based on ViT-B, hence the total model sizes are comparable.

