# OpenReview forum: "Does Visual Pretraining Help End-to-End Reasoning?"
_NeurIPS.cc/2023/Conference — NeurIPS 2023 poster_

### Official Review · Reviewer_PrZv · 2023-06-27

**Soundness:** 3 good
**Presentation:** 4 excellent
**Contribution:** 3 good
**Rating:** 6
**Confidence:** 4

**Summary:**

- The authors explore whether self-supervised visual pretraining can allow end-to-end methods to be more general purpose.
- The paper proposes a self-supervised representation learning framework for this task, based on slot tokens, and evaluates performance on multiple benchmarks.


**Strengths:**

- The paper is well-written, studies an interesting problem, and proposes a satisfying solution.
- The idea of an implicit visual concept learner with self-supervised pre-training, that retains the same benefits as fully symbolic methods, is compelling and will be of interest to the neuro-symbolic & end-to-end learning communities.


**Weaknesses:**

- W1. A key weakness of the paper is that the evaluation datasets are based on CLEVR objects in *synthetic* images. I would like to see experiments on real-world images; I’m not yet convinced that this self-supervised objective with slot tokens would work in natural images that may contain clutter, etc. I am happy to raise my score if authors add evaluation on a real-world dataset.

- W2. I am interested in the performance of using slot-based inputs, instead of explicitly object-centric representations, for neuro-symbolic methods. The proposed methods leverages these softer slot tokens + end-to-end methods; would the change from object-centric input to slot tokens decrease performance of neuro-symbolic methods significantly? Comparison to the NS-CL [1], a neuro-symbolic approach, with slot tokens may be interesting.

[1] The Neuro-Symbolic Concept Learner, ICLR 2019


**Questions:**

See above ^

**Limitations:**

Yes

---

> ### Author Rebuttal · Authors · 2023-08-10
>
> We thank Reviewer PrZv for their thorough review.  We seek to address their listed considerations below:
>
> > **On synthetic evaluation**
>
> We primarily evaluate the reasoning capabilities for IV-CL over established benchmarks for spatiotemporal reasoning, which are largely synthetic due to the difficulty of generating natural-looking video reasoning datasets.  Although these benchmarks are synthetic, they fundamentally succeed at testing reasoning capabilities (such as causal inference, object permanence, etc.) beyond simple recognition.  Furthermore, we outperform other approaches from prior work, which also solely evaluate on such synthetic benchmarks.  We take our reported results as strong preliminary evidence that such models can learn to reason over space and time, and as an encouraging signal that such fundamental reasoning capabilities can transfer over into the future for a natural visual domain. In Table A2 of the supplementary, we also report IV-CL’s performance on the Something-Else benchmark (Materzynska et al., CVPR 2020). Something-Else is a real-world video benchmark focusing on spatiotemporal relationship reasoning, and compositional generalization. IV-CL outperforms the competitive baseline (STIN+OIE+NL) without using object bounding boxes as its supervision. We note that the current state-of-the-art, ORViT (Herzig et al., CVPR 2022), uses ground-truth object bounding boxes during both training and evaluation, and thus is not directly comparable to IV-CL.
>
> > **Using implicit concepts in a neuro-symbolic framework**
>
> We agree that exploring whether softer slot tokens would be beneficial for neuro-symbolic approaches would indeed be interesting. However, this would require translating softer slot tokens into “explicit” concepts as required by the symbolic program, as illustrated in Figure 1(a). A naive quantization approach is likely to cause loss of information, and we thus believe that combining soft concepts with symbolic programs is an interesting but challenging future work.

---

> > ### Comment · Reviewer_PrZv · 2023-08-14
> > **Response to rebuttal**
> >
> > Thanks for the clarifications. I am raising my score due to experiments on the Something-Else benchmark. I would recommend moving these results to the main text, and adding more baselines for comparison.

---

### Official Review · Reviewer_e8rc · 2023-06-28

**Soundness:** 3 good
**Presentation:** 3 good
**Contribution:** 3 good
**Rating:** 6
**Confidence:** 4

**Summary:**

The authors investigate whether unsupervised pre-training yields representations that are suitable for solving reasoning tasks. To that end, they propose a new transformer-based model inspired by masked autoencoders that is pre-trained on CATER. The authors find that the proposed model outperforms supervised pre-training counterparts and it is competitive with respect to object-centric approaches. Interestingly, the representations learned on CATER transfer well to RAVEN, achieving state-of-the-art results. In addition to the main experiments, the authors include multiple ablations and visualizations of the learned slots. Additional experimental details, ablations on the number of slot tokens, visualizations, and pre-training data are provided in the Appendix.

**Strengths:**

Originality
========
* The proposed pre-training scheme is novel to the best of my knowledge
* The finding that representations learned on CATER transfer to RAVEN is novel and interesting

Quality
======
* The proposed method is technically sound
* The model design choices are properly justified with a good number of ablations (6 tables in the main text and one in the appendix).
* Besides comparing with state of the art methods, the authors provide additional baselines such as supervised pretraining, leading to the finding that pre-training on JFT is not significantly better than pre-training on ImageNet.
* The slot visualizations provide additional insights on the inner mechanisms of the learned model.

Clarity
=====
* The text is well-written and easy to read.

Significance
==========
* The fact that unsupervised pre-training on CATER results in representations that compete with dedicated slot-attention methods and transfer to RAVEN matrices is an important finding that will have an impact in the research community

**Weaknesses:**

Originality
========
* Unsupervised pre-training for solving reasoning tasks has already been explored in the past (slot attention) and the proposed method is close to existing methods like MAE. However, I think the originality of this work lies on the empirical evidence obtained from training the proposed method in different ways applied to reasoning tasks.

Quality
======
* The authors make a clear distinction between object-centric two-stage methods using e.g. slot attention, and their proposed approach. However, isn’t the proposed approach also slot-based and two-stage (pre-training + fine-tuning)?

Clarity
=====
* Figure 1 shows a difference between existing slot attention methods (b) and the proposed method (c) in the fact that the proposed method is single-stage. However, does not the method have a pre-training and fine-tuning stage?
* In Figure 2, the slot tokens suddenly appear at the output of the encoders, should not they be in the input too (like the [CLS] tokens)?

Minor
=====
Line 173: reasonign

**Questions:**

* Could you clarify the conceptual difference between unsupervised slot attention methods and the proposed method (see clarity above)?
* Is [A] relevant to your work?

[A] Faulkner, Ryan, and Daniel Zoran. "Solving Reasoning Tasks with a Slot Transformer."

**Limitations:**

The authors include discussion on the limitations in the last section.

---

> ### Author Rebuttal · Authors · 2023-08-10
>
> We thank Reviewer e8rc for their review, and seek to address their listed questions and feedback:
>
> > **On relevant work**
>
> We do agree that the “[A] Solving Reasoning Tasks with a Slot Transformer” is relevant and related to our approach.  We would like to highlight that our IV-CL approach is more general, and utilizes a simpler architecture.  The Slot Transformer work [A] utilizes a slot attention encoder, a temporal transformer, as well as a variational autoencoder in terms of architecture, whereas IV-CL simply utilizes a temporal transformer and an encoder with slot tokens.  Furthermore, the Slot Transformer utilizes two auxiliary losses, whereas IV-CL is a singular reconstruction loss with minimal assumptions during pretraining.  Lastly, the performance that Slot Transformer achieves on CATER (the only overlapping reasoning task) is 62.9% (static camera) / 35.8% (moving camera), whereas IV-CL substantially outperforms it with 70.1% (static) / 59.1% (moving) performance. Please find the results in Table R1 of the rebuttal PDF. We will add the discussion and comparison in the final version.
>
> > **Conceptual difference between Figure 1b (ALOE) and Figure 1c (IV-CL)**
>
> Our definition of “stages” is based on how the input information is processed during inference.  In prior approaches, such as neuro-symbolic (Figure 1a), attention-over-objects (Figure 1b), the input information must be first explicitly organized into objects and their corresponding descriptors (perception stage), before the reasoning stage can be performed. They are hence “two-staged”; the perception model is trained first and frozen, and the reasoning model is trained separately.  The perception model is therefore brittle and inflexible for reasoning, as it is trained without understanding any information about what representations are useful for the downstream reasoning tasks.  In contrast, IV-CL uses end-to-end neural networks without explicitly enforcing any interpretable intermediate representations.  IV-CL can be end-to-end trained to jointly learn the perception model and reasoning model together.
>
> > **On Figure 2**
>
> We agree that the slot tokens should be input as well, but did not visualize them as part of the input for clarity. We have included the revision in the Figure R1 of the rebuttal PDF, will amend this for the final version.

---

> > ### Comment · Reviewer_e8rc · 2023-08-13
> >
> > Thanks for the replies and for running the additional experiments! I will keep the score of 6 since I think *Technically solid, moderate-to-high impact paper, with no major concerns with respect to evaluation, resources, reproducibility, ethical considerations.* does accurately describe your submission.

---

### Official Review · Reviewer_t1ps · 2023-07-06

**Soundness:** 3 good
**Presentation:** 3 good
**Contribution:** 2 fair
**Rating:** 6
**Confidence:** 4

**Summary:**

This paper explores with the need of explicit visual abstraction for visual reasoning tasks.Instead, this paper explores the use of implicit representations learned through a slot attention module.The proposed approach uses slot attention in a frame prediction frame-work that aims to compress visual input information to set of representative tokens/slots. The method is evaluated on CATER, ACRE and RAVEN tasks.

**Strengths:**

* The paper presents an interesting method that seems to generalize across a diverse set of datasets.
* The results on RAVEN are very promising.
* The paper is well written.


**Weaknesses:**

* The proposed method has lot of similarities to prior work: "Learning What and Where: Disentangling Location and Identity Tracking Without Supervision, ICLR 2023", which also uses slot attention to learn implicit representations.

* The above mentioned work achieves 90.7% Top-1 accuracy on CATER, significantly better than the 70.14% reported in this work.

* Therefore, a detained discussion about the similarity and differences to prior work is necessary.

* The proposed method seems to be designed for image sequences. Could the proposed implicit representation be extended to visual reasoning problems which take a single image as input e.g. CLEVR?

* Important details about the evaluation protocol is unclear:: for evaluation on CATER in L242 it is stated that "100 uniformly
sampled frames of stride 3 during evaluation". Why is the full sequence not utilized during evaluation?

* The paper does not report results on the moving camera split of CATER. Why is this the case? Is the method limited to only static camera sequences?

**Questions:**

* An clear explanation of the advantages/limitations with respect to  "Learning What and Where: Disentangling Location and Identity Tracking Without Supervision, ICLR 2023", would be greatly appreciated.

* Details of the evaluation protocol on CATER should be discussed in more detail.

* Details of the applicability of the proposed method to the moving camera split of CATER should be discussed in more detail.


**Limitations:**

Currently, the limitations are not discussed is detail. While evaluation on synthetic data is definitely a limitation, an analysis of failure cases would be helpful.Furthermore, broader societal impacts are also not discussed.

---

> ### Author Rebuttal · Authors · 2023-08-10
>
> We thank Reviewer t1ps for supplying useful feedback in response to our work, as well as relevant prior work, which we address:
>
> > **CATER moving camera performance**
>
> We conduct the moving camera experiments by adopting the same pretrained checkpoint as for the static camera experiments (which was pretrained on videos with static and moving cameras), and using the same hyperparameters for finetuning. On the moving CATER test split, IV-CL achieves 59.1% top-1 accuracy and 89.5% top-5 accuracy. Please also find the results in Table R1 of the rebuttal PDF. The performance is on par with the best ALOE method with auxiliary losses, and better than other baseline methods.  We will update the manuscript with the results of this experiment for the final camera-ready.
>
> > **Comparison with Loci**
>
> The “Learning What and Where: Disentangling Location and Identity Tracking Without Supervision, ICLR 2023" (Loci) paper is indeed relevant to our approach, and demonstrates strong tracking results on CATER. We will discuss Loci as a related work in our final version. We share similar high-level motivations, but highlight the important distinctions: (1) Our work is more general; this is not only reflected in our evaluation, which includes multiple visual reasoning benchmarks beyond CATER, but is also apparent in the design. We do not train IV-CL using specialized task-specific losses, whereas Loci uses three losses specific to tracking and object permanence. IV-CL is general in that it is pretrained using a visual reconstruction task with minimal assumptions, and is finetuned generally on reasoning tasks directly without injecting information about how the task “should be” solved via auxiliary objectives.  (2) Loci assumes static backgrounds and cameras, whereas our approach works with moving cameras and backgrounds. We consider two benchmarks with moving cameras and backgrounds: (1) CATER moving camera split (see our response above), and (2) Something-Else (Materzynska et al., CVPR 2020), a real-world video benchmark focusing on spatiotemporal relationship reasoning and compositional generalization. Table A2 of supplementary shows that IV-CL outperforms the competitive baseline (STIN+OIE+NL) without using object bounding boxes as its supervision. We note that the current state-of-the-art, ORViT (Herzig et al., CVPR 2022), uses ground-truth object bounding boxes during both training and evaluation, and thus is not directly comparable to IV-CL.
>
> > **Extension to single image inputs**
>
> The proposed implicit representation could be extended to visual reasoning problems which take a single image as input, such as CLEVR, by treating the image as a video with a single frame.
>
> > **Frame sampling strategy**
>
> We choose uniformly sampled frames during evaluation to balance memory usage with meaningful frames. Each CATER video has 300 frames. By sampling every 3 frames, the effective video still displays the same actions, but the amount of memory utilized is substantially smaller. Frame sampling during evaluation is a common practice, for example, ALOE sample 80 frames for evaluation.

---

> > ### Comment · Reviewer_t1ps · 2023-08-10
> > **Update**
> >
> > CATER moving camera results improves the paper, will update the rating.
> >
> > However, note that very recent works clearly outperform the proposed approach by a large margin: Look, Remember and Reason: Visual Reasoning with Grounded Rationales, KLR@ICML 2023.

---

> > > ### Author Response · Authors · 2023-08-11
> > > **Thank you for your feedback!**
> > >
> > > Dear Reviewer t1ps,
> > >
> > > Thank you for pointing out this concurrent work which appears at ICML 2023 KLR Workshop! It is indeed very interesting, and we will discuss it in the related work section of our revision.
> > >
> > > Upon a quick reading, we note that there are two main distinctions between IV-CL and the "Look, Remember and Reason": (1) They leverage a large-language model (OPT-125M); (2) Their approach relies on supervised learning of "rationales", which includes object recognition and identification. When such supervision is not provided, their performance is worse than ALOE and IV-CL.
> > >
> > > Best,
> > > The Authors

---

> > > > ### Comment · Reviewer_t1ps · 2023-08-11
> > > > **Update**
> > > >
> > > > Sure, I agree with the author's points.

---

### Official Review · Reviewer_zViR · 2023-07-07

**Soundness:** 3 good
**Presentation:** 3 good
**Contribution:** 2 fair
**Rating:** 5
**Confidence:** 4

**Summary:**

This paper investigates if an end-to-end trained neural network is able to solve challenging visual reasoning tasks. A new representation learning framework named implicit visual concept learner (IV-CL) is proposed. It consists of two main components. First, a single image is compressed into a small set of tokens with a neural network. Second, the slot tokens are provided as context information for other images in the same video, where the goal is to perform video reconstruction via the masked autoencoding objective with the temporal context. The method outperforms traditional supervised pretraining works, and several observations are summarized based on experiment results.

**Strengths:**

The paper is with good motivation and the analysis of current problems of visual reasoning is reasonable and insightful. Different from explicit representations, implicit representations may correspond to part of, or a combination of human-interpretable symbols, or may be discretized and adjusted based on human understanding or model feedback. It has several advantages: free from a pre-defined concept vocabulary or concept classifiers, and avoids early loss of information.

Thus, the paper proposes a new framework named implicit visual concept learner (IV-CL) and summarizes several observations based on experiment results.


**Weaknesses:**

The proposed implicit visual concept learner (IV-CL) is not a sound solution to the motivation of “implicit representation” of the “explicit” reasoning process, because the model based on self-supervised pretraining is totally an implicit one.

The image is compressed into a small set of tokens, the slot tokens are provided as context information via a temporal transformer network for other images in the same video, and the learning process is supervised by masked autoencoding and video reconstruction.
The explicit information adopted in previous works (e.g., [1,2]) is abandoned. The information includes localized objects, object attributes (e.g., color, shape), and spatial relationships. Reasonable implicit reasoning should imply the information. For example, the visual patches attention for each slot is in correspondence with one piece of information. The visualization in Fig.3 shows some interpretable results obtained from the self-supervised training. However, without any supervision, it is not controllable and stable, and difficult to generalize to more complex visual reasoning applications.

Therefore, method design should be reflected and refined to match the goal proposed in the introduction (L36-39): “Some of the learned implicit representations may be discretized into human-interpretable symbols for the purposes of human understanding of and feedback to the model. Others may correspond to part of or a combination of human-interpretable symbols.”


Alternatively, it is also reasonable to study the effect of pretraining on visual reasoning. However:

1.	The introduction starts with the problem of existing explicit reasoning (L33-44). If the core contribution is a new pre-training paradigm, the demonstration should be adjusted.

2.	More importantly, the paper is built upon small, simplified synthetic data, which have a huge gap with practical visual reasoning applications, e.g., VQA in real scenes with diverse visual concepts. Thus, the result and performance of pre-training on these benchmarks are not that convincing and lack generalization. The limitation is discussed in the paper, but I regard it as a crucial problem that should be considered in this paper.


[1] Mao, J., Gan, C., Kohli, P., Tenenbaum, J. B., & Wu, J. (2019). The neuro-symbolic concept learner: Interpreting scenes, words, and sentences from natural supervision. arXiv preprint arXiv:1904.1258

[2] Yi, K., Wu, J., Gan, C., Torralba, A., Kohli, P., & Tenenbaum, J. (2018). Neural-symbolic vqa: Disentangling reasoning from vision and language understanding. Advances in neural information processing systems, 31.


**Questions:**

Discuss the question in the “Weakness” section: Why IV-CL learns "implicit" "concept"? Or why does the pre-training on current datasets make sense?

**Limitations:**

The paper has discussed realistic limitations. However, an important limitation should have been considered (please refer to “Weakness” section).

---

> ### Author Rebuttal · Authors · 2023-08-10
>
> We thank Reviewer zViR for the in-depth, constructive feedback.  Below we seek to address the reviewer’s concerns:
>
> > **On explicit information from supervision**
>
> The reviewer suggests that “without any supervision, [the learned representation] is not controllable and stable, and difficult to generalize to more complex visual reasoning applications”.  In our work, we demonstrate the opposite findings, by comparing our proposed IV-CL framework against a supervised learning paradigm, where slot tokens in the Transformer network are pretrained with supervised visual recognition tasks (e.g. decoding image-level labels or performing object detection). In alignment with the reviewers suggestion, we do present a study on the effect of visual pretraining on visual reasoning in Table 1, which demonstrates the effectiveness of IV-CL over its supervised pretraining counterparts.
>
> > **On the “core contribution”**
>
> The core contribution is not limited to a pretraining paradigm, but also a singular unified framework for learning to reason end-to-end without explicit supervision.  This framework enables the learning of powerful implicit representations, which emerge during visual pretraining and are maintained and refined for reasoning.  The benefit of self-supervised pretraining and finetuning is its generality; compared to explicit representation learning utilized for existing reasoning approaches, our approach requires no explicit human-level supervision.  Therefore, the representations are not fixed and frozen as in prior work, but can flexibly adapt to extract the information useful for the reasoning task at hand during fine-tuning.
>
> > **On synthetic datasets**
>
> Although the datasets are synthetic in nature, the tasks they evaluate require higher-level reasoning capabilities to solve.  The difficulty in these tasks is less about understanding the visual details of the elements in the scene but more about how they relate to and interact with each other.  Therefore, prior works on spatiotemporal reasoning (e.g. OPNet[48], Hopper[69], and ALOE [17]) also perform evaluations on such synthetic benchmarks, which we demonstrate new SOTA for.  However, we do agree that spatiotemporal reasoning on natural datasets would be an interesting and worthwhile endeavor in the future, both in terms of constructing appropriate datasets and exploring our approach on them. In Table A2 of the supplementary, we also report IV-CL’s performance on the Something-Else (Materzynska et al., CVPR 2020) benchmark. Something-Else is a real-world video benchmark focusing on spatiotemporal relationship reasoning, and compositional generalization. IV-CL outperforms the competitive method (STIN+OIE+NL) without using object bounding boxes as its supervision. We note that the current state-of-the-art, ORViT (Herzig et al., CVPR 2022), uses ground-truth object bounding boxes during both training and evaluation, and thus is not directly comparable to IV-CL.
>
> > **On why IV-CL learns “implicit” concepts**
>
> As for what IV-CL learns to encode in the slot tokens, we believe they are implicit concepts.  We believe they are implicit, because no explicit supervision is utilized to learn the concepts.  We believe it is learning concepts through both performant downstream reasoning performance (Tables 4 and 5) and through probing, which verifies object-centric encoding (Figure 3). We will clarify further in the final version.
>
> > **On why pre-training on current datasets make sense**
>
> We believe that the data IV-CL is pretrained on helps it to demonstrate higher-level reasoning capabilities.  Explicitly pretraining on an environment that has CLEVR objects interacting with each other promotes the learning of object-centric representations in a self-supervised way, which also helps it to unlock reasoning capabilities in reasoning tasks that also utilize CLEVR objects.  During finetuning, such implicitly learned object-centric representations are flexibly adapted to the task under our framework; as we demonstrate in Figure 3 (b), after finetuning on ACRE, the slots evolve to capture object relations beyond individual objects, which help it solve the reasoning task at hand.

---

> > ### Comment · Reviewer_zViR · 2023-08-18
> >
> > Thanks for the clarification. Answer 1,2,5 has addressed my questions. I understand that the paper explores achieving higher-level reasoning capabilities while general-purpose pretraining.
> >
> > About answer 3:
> >
> > - The additional result on real-world datasets Something-Else is interesting and solves my concerns.
> > A remaining question about the experimental setting: the reported performance of baseline "STIN + OIE + NL" is actually from "STRG, STIN + OIE + NL". The latter is an ensemble model combining the STRG model. What causes the inconsistency between this paper and Something-Else original paper? Is an ensemble model also used in the reported results of IV-CL?
> >
> > About answer 4:
> >
> > - The item "implicit concepts" seems a little over-claimed.
> > - The intro part  (L35-39) defines the "implicit concepts": (a) a vector-based representation that can be fine-tuned, (b) and support human understanding and feedback. What makes it a "concept" instead of a "transferable representation" is (b) more than (a). The paper and response mainly focus (a). Object-centric encoding (Figure 3) verifies (b), but is it enough to support human understanding and feedback? For example, humans have a thoughtful process to reason out the conclusion, can the process partly reflect in the object-centric encoding (object localization, relationship)?
> > - I understand that it is an open question beyond the scope of this paper. Currently, I suggest re-consideration about the usage and definition of the item "implicit concepts". Also, I am interested in the authors' opinions (optional).

---

> > > ### Author Response · Authors · 2023-08-18
> > > **Authors response**
> > >
> > > Dear reviewer zViR,
> > >
> > > Thank you so much for your inputs! We really appreciate your followup questions and constructive feedback!
> > >
> > > > The reported performance of baseline "STIN + OIE + NL" is actually from "STRG, STIN + OIE + NL".
> > >
> > > This is a good catch! We quoted the best performing method from (Materzynska et al., 2020), and forgot to include "STRG" in the method name, we will fix this in the revision.
> > >
> > > IV-CL result reported in Table A2 is based on a single trained model, and we did not ensemble with STRG or any other models.
> > >
> > > > The item "implicit concepts" seems a little over-claimed.
> > >
> > > We understand and acknowledge this feedback! We agree with the reviewer that our paper mainly focuses on (a), and we use qualitative visualizations to support (b), which show implicit concepts can be optionally discretized into human-interpretable symbols, such as objects or object relations, to support human understanding. We agree with the reviewer that our paper does not investigate how to incorporate human feedback into the learning process, we view it as a desirable property and meaningful future direction. We will include this discussion in the limitations section of the revision.

---

> > > > ### Comment · Reviewer_zViR · 2023-08-18
> > > >
> > > > I thank the authors for the clarification in response and inspiring thoughts in the paper.
> > > > As the concerns are addressed, I raise the score to 5 (from 4).
> > > >
> > > > Possibly, in future work, broader applicability and improved contribution can be achieved with an extensive study on real-world reasoning tasks which are more challenging.

---

> ### Comment · Area_Chair_BYDD · 2023-08-17
>
> Dear Reviewer,
>
> The author has posted their rebuttal, but you have not yet posted your response. Please post your thoughts after reading the rebuttal and other reviews as soon as possible. All reviewers are requested to post this after-rebuttal-response.

---

### Author Rebuttal · Authors · 2023-08-10

We thank the reviewers for their thorough, thoughtful, and constructive comments.  We are happy to hear that the reviewers consistently agree that the proposed work is interesting, well-presented, and presents novel insights.  In particular, the reviewers highlighted that the proposed method is technically sound, that the results on transferring CATER pretraining to RAVEN are both intriguing and novel, and that learning implicit visual concepts is compelling and of interest to the community.

A common comment from the reviewers was the use of synthetic data in our evaluation:

- First, we would like to highlight that it is a common practice (OPNet[48], Hopper[69], ALOE[17]) to evaluate visual reasoning ideas on synthetic datasets. This is due to the difficulty of generating natural-looking video reasoning datasets with sufficiently large scale and accurate annotations. Although these established benchmarks (such as CATER and ACRE) are synthetic, they fundamentally succeed at isolating out and testing reasoning capabilities (such as causal inference, object permanence, etc.) beyond simple visual recognition.

- Second, we have in addition evaluated our method on a real-world dataset Something-Else, in the supplement Table A2. Something-Else is a real-world video benchmark focusing on spatiotemporal relationship reasoning, and compositional generalization. IV-CL outperforms the competitive baseline (STIN+OIE+NL) without using object bounding boxes as its supervision, especially on the "Comp" split which requires compositional generalization.

We would also like to reiterate the core contributions of our work:

- We propose a unified framework for learning to reason end-to-end without explicit supervision.  This framework enables the learning of powerful implicit representations, which emerge during visual pretraining and are maintained and refined for reasoning.

- We demonstrate that a visual pretraining task of masked reconstruction, which makes minimal assumptions about downstream reasoning tasks, is useful in learning powerful representations that capture implicit visual concepts, such as objects and object relations.

- We show via extensive ablation studies and evaluations that our framework can be fine-tuned directly on reasoning tasks without the need for additional task-specific auxiliary losses or inductive biases, and achieves competitive performance or establishes new state-of-the-art on four visual reasoning benchmarks.

We thank the reviewers for their time and careful consideration,

The Authors

---

> ### Author Response · Authors · 2023-08-21
> **Official Comment by the Authors**
>
> Dear reviewers and area chairs,
>
> As the reviewer-author discussion phase is coming to an end, we would like to say thank you, and that we truly appreciate your constructive feedback, which helps substantially improve our submission.
>
> We are glad that our rebuttal and the additional experimental results have addressed the concerns from all reviewers, and we will incorporate the suggested changes (e.g. moving Something-Else results to the main text, clarifying our definition of "implicit concepts", discussing additional related works) in the final version.
>
>
> Best,
>
> The Authors

---

### Decision · Program_Chairs · 2023-09-21

**Decision:**

Accept (poster)

**Comment:**

The paper investigates the role of visual pretraining in end-to-end learning for visual reasoning tasks. It introduces a self-supervised framework that compresses video frames into a set of tokens using a transformer network and evaluates its performance across various benchmarks, notably outperforming traditional methods.

Overall, the reviewers commend the paper for its novel pre-training scheme, promising results on diverse datasets, and technical soundness. The idea of an implicit visual concept learner is particularly well-received for its potential impact on both the neuro-symbolic and end-to-end learning communities. However, there are areas for improvement: the method's originality compared to prior work needs clarification, and the evaluation could be strengthened by including real-world datasets. Additionally, the paper could benefit from revising its alignment with its stated goals and making the implicit representations more interpretable. Some of these weaknesses have been partially addressed in the rebuttal. Given the paper's strengths and its potential for improvement through revisions, the committee leans towards an accept decision.